

# AMOZ

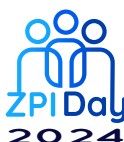

**Autors**: Mikołaj Kwinta · Łukasz Bielawski · Oleksii Adamenko · Andrii Drobitko

**Supervisor:** Mariusz Fraś

## 1  WPROWADZENIE

### 1.1  Tytuł projektu

Aplikacja mobilna do zarządzania zamówieniami dla firm prowadzących sprzedaż hurtową

### 1.2  Autorzy i afiliacje

- **Mikołaj Kwinta** - PM, Analityk, Dokumentalista

- **Łukasz Bielawski** - Programista backend

- **Andrii Drobitko** - Programista frontend

- **Oleksii Adamenko** - Programista frontend

**Abstract**

Celem pracy było stworzenie aplikacji mobilnej służącej do wsparcia małych magazynów lub hurtowni produktów w procesach biznesowych. Podstawowymi funkcjami systemu jest zarządzanie katalogiem produktów oraz pracownikami, kontrolowanie stanów magazynowych oraz realizację zamówień. Główną zaletą projektu jest możliwość dopasowania aplikacji do różnych branż z możliwością indywidualnej oraz elastycznej konfiguracji produktów oraz ich wariantów dopasowanych potrzeb firm. Dostosowanie katalogu produktów do potrzeb odbywa się bezpośrednio przez pracowników bez potrzeby zewnętrznego wdrożenia. Właściciele firm mogą korzystać z systemu aby zarządzać zatrudnienionymi w niej osobami. Projekt ma również na celu uproszczenie procesów magazynowych oraz poprawę efektywności realizacji zamówień oferując firmom narzędzie do zwiększenia produktywności i dostosowanego wsparcia operacyjnego. Dzięki intuicyjnym interfejsom użytkownika, aplikacja jest łatwa do opanowania, co czyni ją atrakcyjnym rozwiązaniem dla małych przedsiębiorstw szukających narzędzi do zarządzania swoją działalnością. W wyniku projektu została zaimplementowana aplikacja z funkcjami pozwalającymi na skuteczne zarządzanie oraz realizację celów wynikających z prowadzenia działalności.

## 2  ROZWINIĘCIE

### 2.1  Wstęp

Współcześnie mniejsze firmy, które chcą rozpocząć działalność oraz zidywidualizować swoją ofertę mają problemy z efektywnym zarządzaniem procesami biznesowymi oraz dynamicznym dopasowaniem katalogu produktów do zmieniających się realiów rynkowych. Brak systemu, który łączy w sobie możliwość zarówno zarządzania ofertą jak i pracownikami oraz monitorowaniem stanów magazynowych prowadzi do nieefektywności takich jak:

- niedobory zapasów na stanach magazynowych

- ręczne śledzenie stanów magazynowych

- nieefektywny sposób zarządzania ofertą

- niedopasowanie do zmieniających się realiów

Co więcej mniejsi przedsiębiorcy potrzebują narzędzia, które pozwoliłoby im w szybki sposób zarówno dostosować obecną ofertę jak i zmienić obszar działalności na taki przynoszący większe zyski. Wiele małych firm nie ma możliwości wdrożyć kosztownych rozwiązań ERP, które znajdują się w ofercie większych korporacji. W związku z powyższym zainstniała potrzeba stworzenia elastycznego i łatwego w użytkowaniu oprogramowania, niewymagającego kosztownych wdrożeń oraz mogącego dopasować się do indywidualnych potrzeb różnych branż.

## 2.2 Prace związane z tematem

### 2.2.1 Technologie

Do implementacji aplikacji AMOZ został zastosowany zestaw technologii, które gwarantują, że system spełnia standardy jakościowe zarówno pod kątem konfigurowalności, bezpieczeństwa, skalowalności jak i przenośności. Aplikacja została postawiona na chmurze Azure aby zagwarantować dostęp do danych oraz bezpieczeństwo użytkowania. Autentykacja Google pozwala na skuteczne zarządzanie użytkownikami systemu oraz wspiera bezpieczeństwo zaimplementowanych rozwiązań.

Architektura trójwarstwowa pozwoliła na łatwiejszą pracę nad aplikacją oraz efektywne rozdzielenie zakresu odpowiedzialności oraz skrócenie ścieżki krytyczej dotyczącej kluczowych zadań projektowych. Aspekty bezpieczeństwa oraz modularności również zostały dzięki temu wzięte pod uwagę umożliwiając łatwą integrację z systemami zewnętrznymi zachowując bezpieczeństwo danych.

Użycie frameworka Jetpack compose zapewnia, że aplikacja jest nastawiona na potencjalny rozwój oraz skalowalność na różne urządzenia w tym na tablety oraz telefony, które mogą być wykorzystywane przez pracowników hurtowni.

Zastosowanie frameworka Spring do backendu pozwaliło na efektywne budowanie szkieletu aplikacji oraz efektywną kominikację opartą na serwisie webowym REST.

Warstwa danych została zaimplementowana na bazie SQL Server, która została sprawnie zintegrowana z Azure SQL Database. Zarówno warstwa logiki aplikacji jak i warstwa danych są dostępne w chmurze z którą się komunikuje aplikacja kliencka.

### 2.2.2 Wyzwania

Utworzenie systemu do zarządzania hurtowniami oraz magazynami w formie aplikacji mobilnej może sprawić wyzwania związane z różnymi etapami wytwarzania oprogramowania. W trakcie implementacji głównymi problemami na które składał się problem były:

- Ustalenie zakresu projektu

- Dopasowanie bazy danych do wymagań biznesowych

- Zarządzanie stanami aplikacji

- Wygasanie tokenów w trakcie korzystania z aplikacji

- Podpięcie procesów pod frontend aplikacji

### 2.2.3 Istniejące konkurencyjne rozwiązania

Na rynku istnieje wiele rozwiązań oferujących systemy zarządzania magazynem hurtowni. Konkurencyjne produkty mają jednak zarówno wady jak i zalety w porównaniu do funkcji oferowanej przez aplikację AMOZ.

#### Rozbudowane systemy do zarządzania magazynem

- **SAP Extended Warehouse Management** - System do zarządzania magazynem dla większych firm z włączonym modułem do zarządzania łańcuchem dostaw

- **Oracle WMS Cloud** - System oparty na rozwiązaniu chmurowym oferujący wsparcie dla logistyki i magazynu.

Systemy te jednak cechują się wysokim kosztem oraz trudnością we wdrożeniu do firmy. Nie oferuje również większej elastyczności, która jest potrzebna do zarządzania procesami w mniejszej firmie. Wymaga również wyszkolonego personelu, aby efektywnie korzystać z tych systemów.

#### Aplikacje mobilne do obsługi magazynów

- **WooCommerce** - Aplikacja mobilna nastawiona na sprzedaż, nie ma jednak modułu odpowiadającego za zarządzanie pracownikami, kategoryzacji produktów oraz stanami magazynowymi. Pozwala jednak na opłatę za produkty w aplikacji.

- **Cashbook** - Aplikacja wspiera analizę wydatków w hurtowni oraz skupia się na generowaniu raportów sprzedażowych. Alikacja AMOZ jest jednak dużo bardzie elastyczna jeśli chodzi o zarządzanie produktami oraz stanami magazynowymi ani nie posiada systemu powiadomień.

- **Sapo Sales Management** - Aplikacja mobilna wspierająca sprzedaż z możliwością integracji z face-bookiem oraz point of sales. Operuje jednak tylko w Tajlandii oraz Wietnamie.

Konkurencja na rynku aplikacji mobilnych istnieje, jednak prezentowane rozwiązania nie pozwalają na zarządzanie ofertą w sposób aż tak elastyczny jak system AMOZ. W prezentowanym systemie poszliśmy na kompromisy pomiędzy rozbudowaniem konkretnych funkcji aplikacji. Głównymi wadami rozwiązań proponowanych na rynku są:

- Koszt wdrożenia

- Brak elastyczności

- Konieczność szkolenia

- Zbyt duża złożoność

## 2.3  Wyniki

Aplikacja AMOZ składa się z funkcji umożliwiających sprawne zarządzanie ofertą, pracownikami, stanami magazynowymi oraz zamówieniami. Rejestracja do systemu AMOZ odbywa się poprzez skorzystanie z autentykacji Google pozwalającej na utrzymanie wysokich standardów bezpieczeństwa użytkowników oraz danych firm. Aby korzystać z systemu, właściciel firmy ma możliwość bezpośrednio ją utworzyć w aplikacji korzystając z przyjaznego interfejsu umożliwiającemu sprawne opisanie jej charakterystyki. Następnie może wysłać zaproszenie do zarejestrowanych użytkowników systemu, którzy muszą je zaakceptować aby zostać jej pracownikami. Właściciel firmy ma następnie podgląd do danych pracowników oraz możliwość modyfikacji ich danych związanych z zatrudnieniem. Zarządzanie katalogiem produktów odbywa się za pomocą mechanizmów kategoryzacji, produktów oraz ich wariantów. Aby poruszać się sprawnie po katalogu, pracownicy mają możliwość spersonalizowania hierarchii kategorii pozwalającej na sprawną nawigację po później skonfigurowanych produktach. Konfiguracja produktów, ich wariantów oraz atrybutów cechuje kilka procesów:

- **Tworzenie product template** - oznacza utworzenie template produktu, w systemie jest wtedy tworzony produkt bez żadnego wariantu

- **Tworzenie zwykłego produktu** - oznacza stworzenie produkt template razem z wariantem podstawowym, który reprezentuje realny produkt w prawdziwym świecie

- **Tworzenie wariantu** - oznacza stworzenie wariantu produktu czyli podpięcie wariantu pod product template

Użytkownicy mają również możliwość skonfigurowania dopasowanych atrybutów produkt template'ów oraz wariantów produktów, które pozwalają na trafne zcharakteryzowanie produktów z różnych branż. Atrybutem produktu może być zarówno pamięć RAM w telefonie dla hurtowni urządzeń elektronicznych jak i gradient papieru ściernego dla hurtowni artykułów budowlanych. Realne produkty istnieją w systemie jako warianty, które mają osobne stany magazynowe, które mogą być aktualizowane przez pracowników firmy lub automatycznie podczas zamówień. Co więcej po osiągnięciu zbyt niskich stanów magazynowych pracownicy mogą dostać powiadomienie informujące ich o zbyt małej liczbie produktów. Stany magazynowe umożliwiają zarówno sprzedaż produktów jako jednostek jak i sprzedaży produktów na wagę, warianty mają jednak osobne stany magazynowe. Proces realizacji zamówień składa się z kilku elementów:

- realizacja zamówienia

- przypisanie klienta B2B lub B2C do zamówienia jeśli jest taka potrzeba

- oznaczenie adresu dostawy jeśli dotyczy

- wystawienie faktury oraz wysyłkę jej wydruku na adres mailowy klienta

Zamówienia mają również różne stany, które odpowiadają za etap realizacji.

Klienci hurtowni mogą również zostać zarejestrowani w systemie aby proces był uproszczony poprzez zapisanie domyślnych adresów dostawy oraz braku potrzeby ponownego wprowadzania danych klienckich.

Aplikacja mobilna AMOZ na systemy Android ma możliwość wprowadzenia różnych trybów (jasny i ciemny) ustawionego w zależności od ustawień telefonu. Wersja językowa jest ustawiona w zależności od wybranego języka na telefonie. Jest ona również skalowalna na różne urządzenia takie jak tablety i telefony mobilne.

# 3 WNIOSKI

## 3.1 Osiągnięcia

W trakcie projektu udało się zrealizować cel w postaci zrealizowania aplikacji mobilnej, która pomaga w zarządzaniu pracownikami, katalogiem produktów, stanami magazynowymi oraz zamówieniami w małych firmach. Rozwiązania zaproponowane przez zespół są łatwe w użyciu ale pozwalające zarówno na dużą elastyczność jak i prostotę działania pozwalającą na uniknięcie kosztownego i długiego wdrażania systemu.

## 3.2 Kierunki rozwoju

Aplikacja pozwala na wybranie wielu kierunków rozwoju, które pozwoliłyby na dodanie lub rozszerzenie niektórych obszarów funkcjonalnych.

**Kierunki rozwoju**

- **Moduł fakturowania** - W przyszłości projekt mógłby zostać rozwijany w kierunku rozwinięcia modułu fakturowego, który pozwoliłby na wystawienie faktur oraz na obsługę różnych typów dokumentów finansowych

- **Logistyka** - Rozwój produktu w kierunku logistyki zarówno pod kątem dostaw do hurtowni, zamawiania z innych oddziałów lub dostaw do klientów

- **Analityka** - Rozwój produktu w kierunku raportów oraz wielowymiarowych analiz pozwalających na podejmowanie korzystniejszych decyzji beznesowych.

- **Platformy** - Rozwój produktu na inne platformy niż tylko android

- **POS** - Utworzenie modułu do prowadzenia sprzedaży internetowej

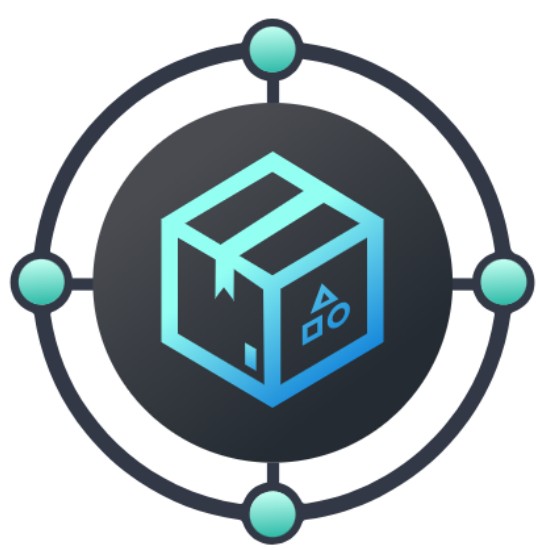

Figure 1: Logo aplikacji

## 3.3 Bibliografia

- Get started with Jetpack Compose [1].

- Spring Boot [8].

- Używanie protokołu OAuth 2.0 na potrzeby dostępu do interfejsów API Google [2].

- App Service documentation [3].

- Azure Blob Storage documentation [4].

- Azure Communication Services documentation [5].

- Azure SQL documentation [6].

- Visual Paradigm Tutorials [7].

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
