# OpenReview forum: "Aplikacja mobilna do zarządzania zamówieniami dla firm prowadzących sprzedaż hurtową"
_pwr.edu.pl/Wrocław_University_of_Science_and_Technology/2024/ZPI_Day — Wrocław University of Science and Technology 2024 ZPI Day Submission_

### Official Review · Reviewer_BmJh · 2024-12-04
**Aplikacja mobilna do zarządzania zamówieniami dla firm prowadzących sprzedaż hurtową - ocena ogólna (w skali 1-5): 2**

**Confidence:** 5
**Significance Of Results:** 5
**Overall Quality:** 2

**Compliance With Template:**

2: Low Quality – The article includes only some of the required sections, and their content is superficial, underdeveloped, or difficult to understand. Some sections are entirely omitted or insufficiently elaborated.

**Description Of Results:**

2: Low Quality – The results are described very superficially and in a general manner. Essential details, usage examples, or evaluations are missing.

**Feedback On Consistency:**

Praca wymaga znacznej poprawy, aby osiągnąć akceptowalny poziom.
Przykłady zauważonych błędów:
- żargon, np. stworzenie (wielokrotnie), podpięcie procesów pod frontend aplikacji, zarządzanie pracownikami (zamiast danymi pracowników), aplikacja mobilna nastawiona na sprzedaż
- brak definicji używanych pojęć, np. małe magazyny, małe przedsiębiorstwo, mniejsza firma, autentykacja Google, możliwość "integracji z facebookiem oraz point of sales"
- niejednoznaczne nazwy, np. template produktu, produkt template, product template, produkt templete'ów
- na początku artykułu: Autors, Supervisor, potem już używany język polski
- w rozdziale Autorzy i afiliacje - wymieniono autorów i ich role w zespole (afiliacja w tym wypadku to wskazanie jednostki realizacji projektu)
- liczne błędy stylistyczne, gramatyczne, interpunkcyjne, literówki, np. "kominikację", "beznesowych", "android", "zcharakteryzowanie", "aplikacja została postawiona na chmurze Azure aby zagwarantować dostęp do danych oraz bezpieczeństwo użytkowania."
- brak odwołania do rysunku 1, brak uzasadnienia po co ten rysunek, podpis rysunku w języku ang. (Figure), zbyt duży rysunek
- pozostawianie "wdów" czyli pojedynczych słów w ostatnim wierszu akapitu
- pozostawianie "sierot" czyli pojedynczej litery na końcu wiersza
- przypisywanie rzeczom cech ludzkich, np. "Aplikacja mobilna AMOZ na systemy Android ma możliwość różnych typów ..."
- niewłaściwe zamieszczone i źle opracowane dwa rozdziały: 3.3. Bibliografia oraz następny (bez numeracji) REFERENCES
- w spisie literatury przy adresach stron internetowych warto podać datę zamieszczenia artykułu i datę dostępu do strony
- brak odwołań w tekście do literatury

**Potential For Development:**

Wskazano na możliwe kierunki rozwoju aplikacji

**Project Nature Evaluation:**

Zespołowy projekt inżynierski

**Technical Language Precision:**

2: Low Quality – The language is partially inappropriate. Significant terminology errors and numerous ambiguities are present. Some sections are imprecise or inconsistent with the expected style of a technical report.

---

### Official Review · Reviewer_QYMf · 2024-12-06
**Aplikacja mobilna do zarządzania zamówieniami - recenzja**

**Confidence:** 5
**Significance Of Results:** 3
**Overall Quality:** 3

**Compliance With Template:**

3: Average Quality – The article includes most of the required sections, but some may be incomplete, written in a general or unclear manner. The content is correct but requires further refinement.

**Description Of Results:**

3: Average Quality – The results are described with moderate detail. Some examples or evaluation elements are present but insufficiently developed or incomplete.

**Feedback On Consistency:**

In general, the work presents a coherent whole, however, in individual chapters the description is often too general and lacks detailed descriptions of the solutions used (e.g. instead of the general "maintaining high security standards" it would be better to list the solutions used for this purpose) or the described techniques or applications. The work somewhat weakly emphasizes those solutions that are key elements of achieving the business goal (e.g. the issue of a cloud solution). The title of the work AMOZ is a complete enigma. An extension or subtitle explaining the nature of the project is necessary.

**Potential For Development:**

The paper indicates possible directions for the development of the application. They are formulated somewhat laconic and it is not entirely clear how they relate to the important goals set for the project.

**Project Nature Evaluation:**

The project meets the characteristics of an engineering work. The application was implemented using proven technologies and providing appropriate functionalities. However, the description lacks a deeper analysis behind the choices made. Unfortunately, the literature is very improperly provided (it is practically unknown what these sources are) and references are virtually non-existent in the text. The undoubted usefulness of the application not well emphasized in terms of its specific and unique features. The techniques used to implement the product were described quite briefly.

**Technical Language Precision:**

4: High Quality – The language is appropriate for a technical report. Terminology is used correctly, and statements are precise, with only minor shortcomings that do not affect the overall clarity.

---

### Official Review · Reviewer_KGLg · 2024-12-08
**The article deals with an interesting issue, but the description indicates a general knowledge of the processes to be supported by the application.**

**Confidence:** 3
**Significance Of Results:** 3
**Overall Quality:** 3

**Compliance With Template:**

3: Average Quality – The article includes most of the required sections, but some may be incomplete, written in a general or unclear manner. The content is correct but requires further refinement.

**Description Of Results:**

3: Average Quality – The results are described with moderate detail. Some examples or evaluation elements are present but insufficiently developed or incomplete.

**Feedback On Consistency:**

The analysis of problems and presentation of results is very general. There is no logical link between the indicated functions of the application. The authors indicate that the application will be used for personnel management and then only describe that employees will be able to register while the owner will be able to enter their employment data. The same applies to the other functionalities. The indicated developments are described at a very high level of generality. The quality of the content presented is negatively affected by spelling, stylistic and punctuation errors

**Potential For Development:**

The article indicates possible directions for the development of applications. However, it is necessary to think them through and define them precisely. Using big terms like ‘Logistics’ for simple tasks like ‘delivery’ is misleading to the potential customer (user).

**Project Nature Evaluation:**

The design shows the characteristics of an engineering paper, but the scope of functionality of the proposed application needs to be rethought and refined.

**Technical Language Precision:**

3: Average Quality – The language is mostly appropriate but may contain minor terminological or stylistic errors. Some statements might lack precision or require improvement for better readability.

---

### Decision · Program_Chairs · 2024-12-10

Accept (Poster)